# EGFR and KRAS Mutations in the Non-Tumoral Lung. Prognosis in Patients with Adenocarcinoma

**DOI:** 10.3390/jcm8040529

**Published:** 2019-04-17

**Authors:** Roberto Chalela, Beatriz Bellosillo, Víctor Curull, Raquel Longarón, Sergi Pascual-Guardia, Diana Badenes-Bonet, Edurne Arriola, Albert Sánchez-Font, Lara Pijuan, Joaquim Gea

**Affiliations:** 1Respiratory Medicine Department, Hospital del Mar, 08003 Barcelona, Spain; VCurull@parcdesalutmar.cat (V.C.); SPascual@parcdesalutmar.cat (S.P.-G.); DBadenes@parcdesalutmar.cat (D.B.-B.); ASanchezF@parcdesalutmar.cat (A.S.-F.); JGea@parcdesalutmar.cat (J.G.); 2IMIM (Hospital del Mar Medical Research Institute), 08003 Barcelona, Spain; BBellosillo@parcdesalutmar.cat (B.B.); Earriola@parcdesalutmar.cat (E.A.); 3School of Health & Life Sciences, Universitat Pompeu Fabra, 08003 Barcelona, Spain; 4CIBER de Enfermedades Respiratorias (CibeRes), Instituto de Salud Carlos III, 28029 Madrid, Spain; 5Department of Pathology, Hospital del Mar, 08003 Barcelona, Spain; RLongaron@parcdesalutmar.cat (R.L.); LPijuan@parcdesalutmar.cat (L.P.); 6Universitat Autònoma de Barcelona, 08003 Barcelona, Spain; 7Department of Oncology, Hospital del Mar, 08003 Barcelona, Spain

**Keywords:** Adenocarcinoma, Mutations, *EGFR*, *KRAS*, Prognosis

## Abstract

Tumor recurrence is frequent and survival rates remain extremely low in lung adenocarcinoma (ADC). We hypothesize that carcinogenic factors will promote loco-regional modifications not only in the future tumor, but throughout the exposed lung. Objective: To analyze whether the most prevalent mutations observed in ADC can also be observed in the non-neoplastic lung tissue, as well as the short-term prognosis implications of this finding. Methods: Non-tumoral lung parenchyma specimens obtained during surgery from 47 patients with *EGFR* and/or *KRAS* abnormalities in their ADC tumors underwent similar genomic testing. Short-term outcomes were also recorded. Results: The same mutations were present in the tumor and the histologically normal tissue in 21.3% of patients (SM group). Although local recurrences were similar in both groups, distant metastases were more frequent in the former (60 vs. 5.4%, *p* < 0.001). Moreover, SM patients showed lower time-to-progression (8.5 vs. 11.7 months, *p* < 0.001) and disease-free survival (8.5 vs. 11.2 months, *p* < 0.001). COX regression showed a higher risk of progression or death (DFS) in the SM group (HR 5.94, *p* < 0.01]. Similar results were observed when adjusting for potential confounding variables. Conclusions: These results confirm that genetic changes are present in the apparently normal lung in many ADC patients, and this finding has prognostic implications.

## 1. Introduction

Lung cancer is the second most frequently diagnosed tumor and the leading cause of cancer-related deaths worldwide [1,2]. There are two main histological types of lung neoplasms, non-small-cell lung cancer and small-cell lung cancer. The former group includes lung adenocarcinoma (ADC), which is the most prevalent subtype of lung neoplasms [3]. Despite major efforts in smoking prevention policies and early detection of lung cancer, as well as advances in research, including new therapies, the overall 1-to 5-year survival rates remain extremely low. Although surgical resection is the first-line treatment for early-stage ADC, tumor recurrence is still the most common cause of morbidity in these patients [4,5,6,7] Moreover, in those patients who underwent ‘curative surgery’ the 5-year survival is less than 60% [8,9,10].

Genetic and epigenetic changes occur during carcinogenesis. In fact, all the malignant cells present DNA modifications at some point during this process and/or the proliferation step. These DNA acquired changes are known as somatic genomic alterations, being divided into “passenger mutations”, those which are supposed not to be related with the development of cancer; and “driver mutations”, if they are directly involved in carcinogenesis [11,12,13]. Both ADC and squamous cell lung carcinoma have a high mutational burden when compared with other cancers. Moreover, studies published by the Cancer Genome Atlas (TCGA) have found driver mutations in more than 75% of the lung ADC, identifying 35% of such mutations in oncogene TP53. This situation overlaps with oncogenic driver alterations that have potential therapeutic implications such as mutations occurring in *KRAS*, *EGFR*, *BRAF*, *MET*, *ERBB2* and gene fusions taking place in *ALK*, *ROS1* or *RET* [14,15]. Nevertheless, up till now the International Association for the Study of Lung Cancer (IASLC)/American Thoracic Society (ATS)/European Respiratory Society (ERS) panel of experts recommended molecular testing only for *EGFR* mutation in advanced ADC. However, recent evidence from positive clinical trials using genome-guided therapies suggest the convenience of extending the search [16,17,18]. In fact, the updated Clinical Practice Guidelines in Oncology also recommends a broader molecular profiling since this is important to identify rare mutations (for which therapeutic drugs may already be available)^18^. Furthermore, the presence of driver mutations in the tumors of patients with lung ADC may not only be important for use in specific therapies but also for their prognostic implications in the early-stage disease [19,20,21]. Our hypothesis was that most probably carcinogenic factors will promote loco-regional modifications, not only in the future tumor, but throughout the exposed tissues. Accordingly, the aims of the present study were to identify whether the most prevalent mutations observed in lung ADC were also present in the histologically non-tumoral lung tissue (NTL) of the same patient and, if this was the case, to assess their potential usefulness as prognostic markers.

## 2. Experimental Section

### 2.1. Patients

From 2011 to 2016, 625 patients with lung ADC were diagnosed in our center, a tertiary teaching-hospital. One hundred and sixty-nine (26.8%) of these patients were candidates for curative resection. All tumor samples obtained during surgery underwent genomic testing to detect *EGFR* and *KRAS* for therapeutic purposes. As a result, *EGFR* and/or *KRAS* mutations were identified in 57 patients. These subjects were initially included in the present study and blocks of their normal lung parenchyma were processed to extract DNA. Normal lung parenchyma was defined as histologically normal tissue as assessed by two expert lung pathologists. It was always taken from a peripheral area of the surgical piece at least 2 cm from the tumor. Finally, viable DNA was obtained in 47 of these patients, being tested with highly sensitive-specific genomic techniques to identify the same mutation previously found in the tumor. The quality of DNA from the other patients was considered of low quality [22,23]. The clinical outcomes of patients included in the study were recorded for at least one year following surgery, although they now remain to be followed up. The study was designed and carried out in accordance with the ethical guidelines of the Declaration of Helsinki and European legislation, and the procedure was approved by our Ethics Committee. Informed consent was obtained from all individuals or their closest relatives.

### 2.2. Data

General and clinical data were collected for months 1, 2, 6 and 12 following thoracic surgery. These data included anthropometric and sociodemographic characteristics, past medical history, smoke status, lung function tests, pre-operative blood analysis, histological classification of ADC^3^, determination of disease stages based on the tumor, node, and metastases (TNM) classification (IASLC, 8th edition), and the pathological characteristics of the tumor. All but one patient, who died one month following surgery, completed the one-year follow up in our own outpatient clinic, with at least two thoracic and upper abdominal computed tomography (CT) scans performed in months 6 and 12. Positron emission tomography (PET)–CT scan, nuclear medicine techniques or brain magnetic resonance were done if suspicious symptoms occurred. The mortality was accessed through local health system reports and phone calls.

### 2.3. Tumor DNA Extraction and Sequencing

DNA was extracted from tumoral sections of each sample with the commercially available QIAamp DNA Mini kit (Qiagen, Hilden, Germany). The EGFR mutational status were analyzed by real-time PCR using the TheraScreen EGFR RGQ PCR kit (Qiagen, Hilden, Germany), a highly sensitive assay based on Scorpions^®^ real-time PCR technology (Qiagen, Hilden, Germany), and mutation specific ARMS^®^ (Qiagen, Hilden, Germany)primers that detect 29 different somatic mutations in the gene. In addition, 18, 19, 20 and 21 exons of the EGFR gene, as well as exon 2 of the *KRAS* gene were analysed in all cases by Sanger sequencing, using BigDye v3.1 (Applied Biosystems, Foster City, CA, USA), being assessed on the 3500DX Genetic Analyzer (Applied Biosystems, Foster City, CA, USA).

### 2.4. DNA Extraction and Allele-Specific PCR in the NTL Tissue

After the identification of the non-tumoral tissue of patients with the diagnosis of *EGFR* or *KRAS*-mutant lung adenocarcinoma, DNA was extracted from 2 15 μm sections of using the QIAamp DNA Mini kit (Qiagen, Hilden, Germany). Mutational analysis was performed in this case using competitive allele-specific TaqMan PCR (CAST-PCR, Applied Biosystems, 4465804, Foster City, CA, USA). The following individual assays were used: EGFR exon 19 deletions-Hs00000228_mu; EGFR p.L858R-Hs00000102_mu; EGFR p.T790M-Hs00000106_mu; G719A-Hs00000104_mu; KRAS p.G12C-Hs00000113_mu; KRAS p.G12V–Hs00000119_mu; KRAS p.G12D-Hs00000121_mu; KRAS p.G12A-Hs00000123_mu; KRAS p.G12R–Hs00000117_mu; and KRAS p.G13C-Hs00000125_mu.

### 2.5. Digital PCR

In addition, to confirm the results obtained by TaqMan PCR (CAST-PCR, Hilden, Germany) and to assess the percentage of mutated copies, digital PCR was used in the non-tumoral tissue. For this technique the sample was partitioned to the level of single molecules and then the amplification was performed.

### 2.6. Definitions

The tumor stage was calculated with postoperative findings (tumor size, lymph node involvement and metastasis). Diagnosis of recurrence was established when local progression, lymph node involvement or distant metastases were detected following lung resection, in the follow up period. Local recurrence was defined as new cancer involvement in the same hemithorax and/or mediastinum, whereas metastasis was defined as involvement in other locations. Time to progression (TTP) was defined as the period between surgical treatment and recurrence. The Disease-Free Survival (DFS) was defined as the period of time after treatment that the patient survived with no evidence of cancer progression. After the identification of patients with *EGFR* or *KRAS* mutations both in the tumor and the non-tumoral lung tissue, two groups were defined to analyze the results: 1. The same-mutation group (SM-Group): Cases with the same driver-mutation in the lung adenocarcinoma and the non-tumoral tissue; and 2. The Non-SM Group: Cases with the presence of a driver-mutation in the lung adenocarcinoma but with wild-type status in the non-tumoral tissue.

Non-tumoral lung was defined, as recommended by the World Health Organization (WHO) tumor classification, as the absence of malignant cells upon light microscopy using routine hematoxylin and eosin. These samples were analyzed by two expert thoracic histopathologists and additional immunohistochemical staining could be used to rule out malignancy, as is done in routine clinical practice [3].

### 2.7. Statistical Analysis

While categorical variables are described as frequencies and percentages, continuous variables are expressed as mean ± standard deviation. Pearson’s Chi-Square or Fisher exact tests were used as appropriate to compare categorical variables between groups. The non-parametric Mann-Whitney U test was used to assess differences between groups. Both TTP and DFS (which includes both progression and death) were investigated using the Kaplan–Meier method. Finally, the log-rank test was used to make comparisons between Kaplan-Meier outcomes, and the Cox proportional hazard model was employed for univariate and multivariate survival analyses. *p* values ≤ 0.05 were considered statistically significant. Analyses were performed with SPSS 21.0.

## 3. Results

All patients were Caucasian, their mean age was 66 years (range, 43–83 years), and predominantly male (55.3%). The 47 cases included in the study were divided into two groups, the SM Group and the Non-SM Group to analyze the results. Their main clinical and functional characteristics are shown in Table 1. The vast majority of patients included in the study had pre-operative N0 status, and therefore the first-line treatment was surgery without neo-adjuvant therapy in all but two patients. Surgical procedures were lobectomy (74.5%), bilobectomy (10.6%) and segmentectomy. The treatment after surgery was performed according to the post-operative stage, mutational status and PD1/PDL1 expression in the tumor as recommended by international guidelines and local protocols.

### 3.1. Driver Mutations in the Tumor

The tumor was *EGFR*+ in a group of 24 patients and *KRAS+* in another set of 24 subjects (Table 2 and Figure 1). Only one patient carried two mutations (which are usually mutually exclusive), in *EGFR* (Ex.18 G719A) and *KRAS* (Gly13Cys), and four more carried two different mutations in their *EGFR* gene. The most frequent mutations detected in *KRAS* positive patients were the substitution of glycine by cytosine in codon 12 (Gly12Cys) (50% of the cases), followed by the substitution of valine (Gly12Val) and aspartic acid (Gly12Asp) (both in 16.6% of the subjects). The most frequent mutations found in *EGFR* in turn were the E746_A750 (exon 19) deletion (41.6%), followed by the activation-mutation of L858R (substitution of arginine for leucine at codon 858 in exon 21) (29.2% of the cases).

### 3.2. Mutations in the NTL

The same mutation observed in the tumor was observed in the NTL of 10 patients (21.3%) of the SM group, with similar distribution for EGFR and KRAS mutations (50% of the patients each). In one patient, who carried two *EGFR* alterations (Ex.18 G719A and Ex.20 T790M) in the tumor, both mutations were also identified in the normal parenchyma. Hematoxylin-eosin staining of non-tumoral lung tissue and lung adenocarcinoma of two patients of the SM group are shown in Figure 2.

Quantification of mutated copies by Digital PCR showed a mean of 0.20% (minimum 0.02, and maximum 0.50%) with respect to total copies of the gene in those cases where *EGFR* mutation was detected. In the cases where *KRAS* mutation was formerly observed, a mean of 0.08% (0.02–0.17%) of mutated with respect to total copies was identified.

### 3.3. Recurrence and Survival

Data from both groups are shown in Table 1 and Table 3. They only differed in the smoking index, DLco and SUV (PET).

One patient died in the first month following surgery. During the 12 months follow-up, two more patients died, one in each group. However, SM patients presented a much higher recurrence rate compared with non-SM subjects. The pattern of recurrence and the organ sites affected by distance metastases can be observed in Table 3. Surprisingly, local recurrence only occurred in one patient from the non-SM group, meanwhile distant metastases occurred in 60% vs. 5.4% in SM and non-SM groups, respectively. The brain was the most prevalent organ affected by distant metastases, being observed in almost all SM patients, whereas two or more organs were affected in around a third of subjects in the same group. No differences were found in either recurrence or survival when SM patients with *EGFR* mutations were compared with those with *KRAS* alterations.

Kaplan-Meier analysis (Figure 3) showed that TTP was lower in the SM than in the non-SM group (8.5 months, (95% confidence interval (CI), 6.4–10.5) vs. 11.7 months (95% CI, 11.3–12); and the DFS was also significantly poorer in the former (8.5 (95% CI, 6.4–10.6) vs. 11.2 months (95% CI, 10.5–11.9). In other words, only 40% of SM patients were alive and recurrence free at the first year, while they were 86.5% in the non-SM group (*p* < 0.01).

COX regression showed a higher progression or death risk (DFS) in the SM group (hazard ratio (HR) 5.94 (95% CI, 1.7–19.6, *p* < 0.01)). In the multivariate analysis, when adjusting for sex, age, smoking status, lymphovascular invasion and postoperative pathologic stage, SM patients maintain a significantly higher risk of progression (HR, 12.1 (95% CI: 2.6–56.8), *p* = 0.001) and a significantly worse DFS than non-SM subjects (HR, 7.1 (95%CI 1.9–26.5), *p* < 0.01).

## 4. Discussion

This is the first study demonstrating the presence of driver mutations in the non-neoplastic lung tissue of patients with lung ADC. Moreover, our study also shows that those patients with similar mutations in the tumor and the non-neoplastic lung parenchyma have more precocious recurrences and less DFS at the first year following surgery. Our findings can also have implications in the conception of the pathophysiology of carcinogenic processes occurring both in the lung and at distance (metastasis).

ADC is currently the most frequent lung tumor in developed countries, and in many cases (especially in women) it is not directly related to tobacco smoking but to other factors such as environmental or labor exposure, and/or a facilitating genetic background. Unfortunately, mortality due to ADC remains intolerably high even despite advances achieved in recent years. This is sadly true, even for patients in the initial stages of the disease, who are candidates for surgical treatment for supposedly curative purposes. It is therefore necessary to improve our knowledge of the pathophysiology of the occurrence, growth and dissemination of ADC. The most widely accepted theory is that the onset of tumors depends both on genetic background, exposure to carcinogenic factors and an appropriate cellular microenvironment [11]. The current knowledge about the process of cancerization include that tissues exposed to carcinogenic factors will gain a large variety of loco-regional modifications, with often little or no evidence in histological analyses. This certainly appears to be the case in our SM patients. Following the current carcinogenesis theory, these modifications (‘hallmarks of cancer’) slowly accumulate and facilitate the onset, evolution and progression of the tumor. Of relevance is that some of these modifications are epigenetic, so they could also have implications for descendants. In particular, in pulmonary ADC up to 15 methylated regions, which are absent in normal lung tissue, have been identified [24,25].

The mutations observed in the tumor in the present study do not differ substantially from those previously described for Caucasian patients exposed to tobacco smoking [19,26,27,28]. However, these genetic changes were present both in the tumor and the NTL parenchyma. This interesting finding is in close agreement with the aforementioned theory of carcinogenesis. According to this theory, a reasonable interpretation of the present results would be that following exposure to carcinogenic agents, different areas of the target organ, in this case the lung, would develop genetic abnormalities. In a certain moment of the process, these progressive changes would facilitate the onset of cancer in one of these zones, probably in relation with local microenvironmental factors. This could also imply that with enough exposure time, other areas could accumulate enough ‘hallmarks of cancer’ to further develop a tumor. The earlier presence of distant metastases in the group of patients with mutations found in non-tumoral areas may be interpreted either as the arrival of tumoral cells from the primary tumor to these distant organs or, alternatively, the nesting of non-neoplastic cells coming from other parts of the lung. These cells that are already carrying a mutational charge may progress in their new location, potentially leading to a new tumor. The latter possibility, still somewhat speculative, would have ontological consequences regarding conceptions of ‘recurrence’ or even of ‘metastasis’ from the ‘primitive tumor’, which should then be considered a new tumor. Much more relevant than this otherwise semantic point, is that our results also evidence another interesting finding: those patients with similar mutations in the tumor and the NTL parenchyma disclose more metastasis and a lower DFS than those patients without. This may suggest that in the future, the use of a more aggressive approach in the follow up of these patients could be a valid attitude, and the use of adjuvant therapy should be proposed to the subset of these patients in order to prolong DFS. However, caution is required when making conclusions about a worst prognosis for these patients, since they are based on a single center study with a relatively small sample size. Thus, a wide multicentric study is probably required to confirm our results.

An interesting point is that DLco, a typical functional marker of emphysema, was lower in SM patients. It is known that emphysema is associated with a greater facility for the development of lung cancer, even in the absence of COPD criteria [29]. Although no morphological data on the degree of emphysema are available in the present study, it could be speculated with the ‘facilitation’ of the emphysematous cellular microenvironment for the occurrence of mutations in different locations of the lung, which subsequently can progress to cancer.

### 4.1. Alternative Study Designs

When considering methodological alternatives to confirm our findings, multiple difficulties arise that reaffirm the importance of a real-life human model. An animal model would require a tumorogenic exposure similar to that of patients (i.e., tobacco smoking) for a long period of time and in a multitude of wild-type animals, since only a minimal part would develop an adenocarcinoma, and only a percentage would develop the genetic changes that are included in our results. The alternative use of genetically manipulated animals would distort the results since they do not reproduce the clinical situation observable in humans. Nor does it seem logical to use cell cultures for the purposes of the present study since it is already known that exposure to tobacco promotes deleterious mutations in the exposed cells. In both cases (animal models and cell cultures), new data would not be provided in relation to one of our most relevant findings, the association between mutations identified in the ‘non-tumoral tissue’ and patients’ short-term prognosis.

### 4.2. Potential Limitations of the Study

Lung samples were always obtained during resective surgery, and they were therefore always from the same lobe as the cancer. Even though non-tumoral lung tissue has always been obtained from the area furthest from the primary tumor, we cannot know the real situation in the other areas of the lung. Therefore, the present results are not necessarily generalizable to the whole parenchyma. However, this is likely since the genetic background was equivalent for the entire organ and it seems reasonable to assume a similar exposure to inhaled carcinogens. Moreover, there is no alternative for this potential limitation. Even though, theoretically, it would be ideal to demonstrate that the driver mutations can occur in the contralateral lung, the procedure for obtaining this tissue is ethically unacceptable as it would require invasive and potentially dangerous techniques for the patients. An alternative would be to obtain material from a distant lobe of the same lung during surgery, but this also involves serious ethical concerns.

The sample size of the present study is relatively small to draw definitive conclusions regarding the medium-term prognosis of our patients and for this reason a larger study with a multicenter approach should be necessary in the future. However, it is sufficient to demonstrate the main objective of the study which was to determine the presence of driver-mutations in non-tumor lung tissue and is similar to previous published studies in this field [30,31].

With regards to the potential ‘contamination’ of the non-tumoral tissue from the original tumor, it seems to be ruled out since the NTL analyzed was always taken farthest away from the ADC, and because of the absence of lymphovascular invasion in all the patients. Moreover, although the presence of circulating tumor cells, which may potentially have contributed to the positive results, cannot be totally ruled out, it is unlikely. The presence of these cells or circulating tumoral DNA is strongly dependent on the tumor size and extension, and the present cohort was composed of patients with early-stage lung cancer. Moreover, at this stage, the detection of *EGFR* mutations in blood is low and detection of *KRAS* has not been yet reported.

Finally, our study still provides no data on medium or long-term survival, since unfortunately, only one-year follow-up is already available. This is the main reason for having used DFS as one of the main outcome variables, since it includes both recurrence and survival dimensions. However, patients are being followed up now, and the authors will be able to show the long-term survival of both groups in the near future.

## 5. Conclusions

The conclusions of the present study are that in a relatively high proportion of patients with *EGFR* and/or *KRAS* mutations in their ADC, these genetic abnormalities are also present in their histologically normal lung tissue. Apart from the possible consequences that this finding may have on our conception of both the carcinogenesis process and the extension of a primary tumor to other organs, the presence of these extratumoral mutations seems to be associated with a worse short-term prognosis. The latter may have clinical consequences in the future follow-up of this patients.

## Figures and Tables

**Figure 1 jcm-08-00529-f001:**
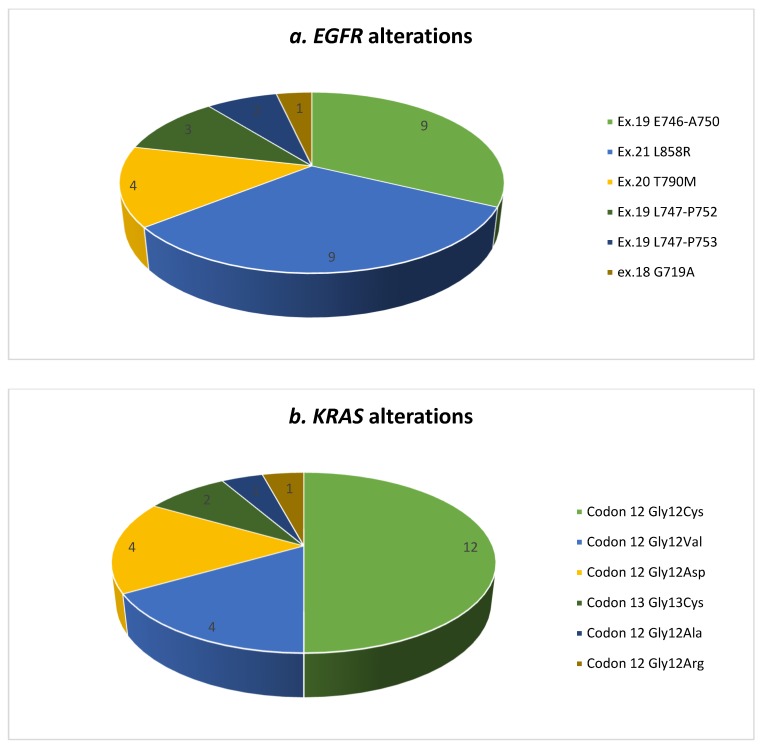
Graphical representation of the mutation type identified for both EGFR and KRAS. (**a**) EGFR alterations identified; (**b**) KRAS alterations identified.

**Figure 2 jcm-08-00529-f002:**
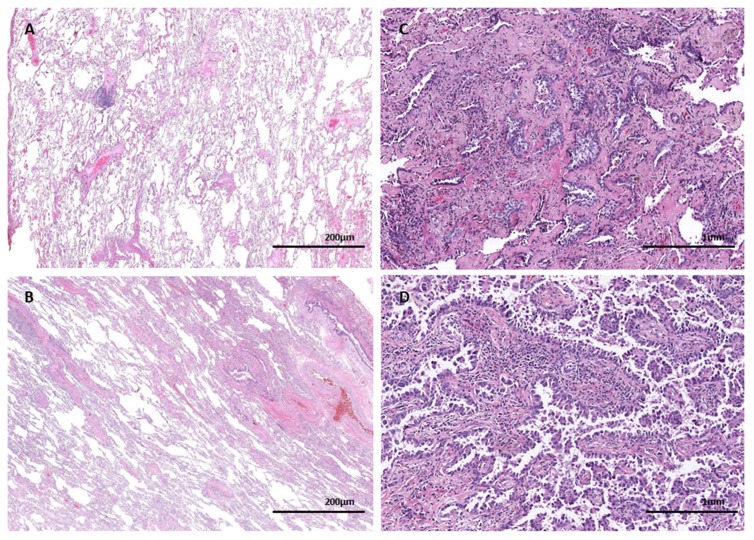
Hematoxylin-eosin staining of two cases with mutations in both non-tumoral lung tissue and lung adenocarcinoma: (**A**) non-tumoral lung sample of patient with *EGFR* mutation (2×) and (**C**) lung adenocarcinoma sample of the same patient (10×); (**B**) non-tumoral lung sample of patient with *KRAS* mutation (2×) and (**D**) lung adenocarcinoma sample of the same patient (10×).

**Figure 3 jcm-08-00529-f003:**
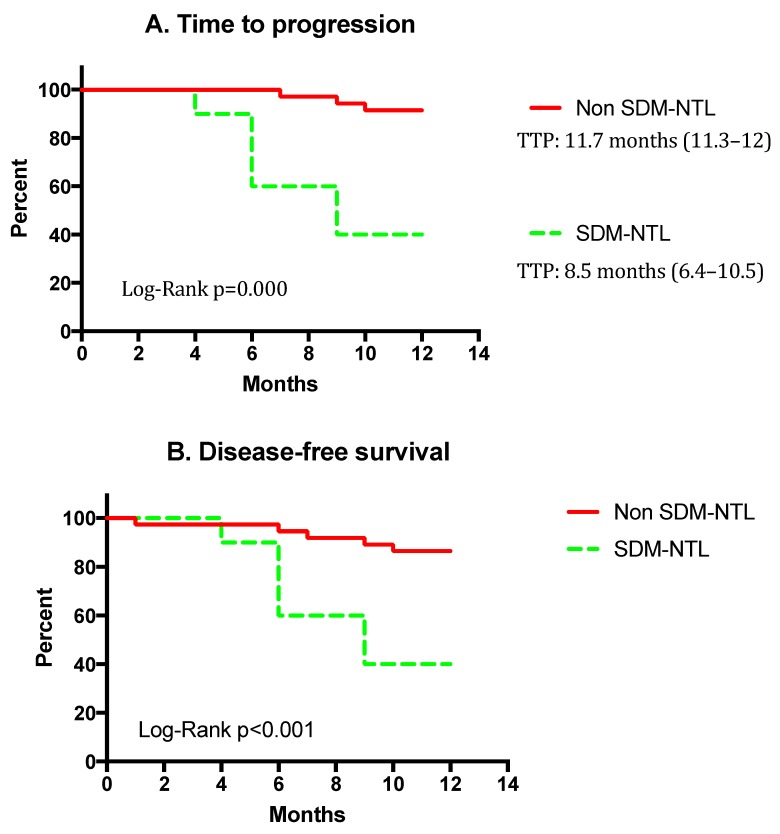
Kaplan Meier curves for (**A**) time to progression, and (**B**) disease free survival time. Abbreviations: SM, genetic alterations (*EGFR* and/or *KRAS*) observed both in the tumor and the non-tumoral lung parenchyma. Non-SM, genetic alterations shown only by the tumor.

**Table 1 jcm-08-00529-t001:** Comparisons of Baseline characteristics in SM and non-SM groups.

	SM *n* = 10	Non-SM *n* = 37	*p* Value
Age, mean (SD), yrs.	62.9 (8.7)	66.9 (9.9)	0.25
Smoking status, current or former, *n* (%)	6 (60)	24 (64.9)	0.77
Smoking index, mean (SD), pack-year	**62.1 (27)**	**41.2 (20)**	**0.04**
Sex, *n* (%)			
Male	5 (50)	21 (56.8)	0.70
Female	5 (50)	16 (43.2)	
Comorbidities, *n* (%)			
Diabetes mellitus	1 (10)	4 (10.8)	0.94
Chronic kidney disease	1 (10)	0 (0)	0.06
Cardiovascular disease	2 (20)	5 (13.5)	0.60
Dyslipidemia	2 (20)	18 (48.6)	0.10
Hypertension	4 (40)	21 (56.8)	0.34
COPD	3 (30)	8 (21.6)	0.57
Asthma	1 (10)	2 (5.4)	0.59
Previous cancer	2 (20)	10 (27)	0.65
Lung function tests, mean (SD)			
FEV_1_, % ref.	78.3 (19.1)	79.1 (20)	0.92
FVC, % ref.	85.2 (17.9)	83 (17.4)	0.75
TLC, % ref.	98.2 (13.6)	96.9 (14.7)	0.82
RV/TLC, %	48.5 (9.3)	47.4 (12.4)	0.77
DLCO, % ref.	**66.7 (15.1)**	**82.9 (19.5)**	**0.04**
*Karnofsky Performance Scale, mean (SD)*	88 (6.3)	88.6 (3.4)	0.66
*Pre-operative tumor characteristics*			
SUV by PET, mean (SD), cm	**9.0 (5.7)**	**4.4 (4.7)**	**0.01**
T (tumor size), mean (SD), cm	3.3 (16)	2.6 (17)	0.75
N (nodal infiltration), *n* (%)	1 (10)	1 (2.7)	0.31
M (metastasis), *n* (%)	0 (0)	0 (0)	
*Post-operative Stage Groups, *n* (%)*			0.43
I	6 (60)	29 (78.4)	0.21
II	3 (30)	5 (13.5)	0.34
III	1 (10)	3 (8.1)	1.00
IV	0 (0)	0 (0)	--
*Post-operative Treatment, n (%)*			
Chemoradiotherapy	4 (40)	10 (27)	0.42
Radiotherapy	4 (40)	7 (18.9)	0.16
Genomic guided treatment or immunotherapy	**4 (40)**	**1 (2.7)**	**0.001**

*Abbreviations:* SM, same mutations in the tumor and the non-tumoral parenchyma; SD, standard deviation; COPD, chronic obstructive pulmonary disease; FEV_1_, forced expiratory volume in the first second; FVC, forced vital capacity; TLC, total lung capacity; RV, residual volume; DLco, transfer coefficient for CO; SUV, standardized uptake value; PET, positron emission tomography.

**Table 2 jcm-08-00529-t002:** Details on Driver Mutations found in the Lung Tumor.

*EGFR* Mutations (24 Patients *; Total Mutations = 28)				
Site	Alteration	Amino Acid Variation	Patients, *n*	Group (*n*+)
Exon 19	Deletion	E746-A750	9	SM (3)
Exon 21	Substitution	L858R	9	SM (1)
Exon 20	Substitution	T790M	4	SM (3)
Exon 19	Deletion	L747-P752	3	SM (1)
Exon 19	Deletion	L747-P753	2	Non-SM
Exon 18	Substitution	G719A	1	Non-SM
*KRAS* mutations (24 patients *; total mutations = 24)				
Site	Alteration	Amino acid variation	Patients, *n*	Group
Codon 12	Substitution	Gly12Cys	12	SM (3)
Codon 12	Substitution	Gly12Val	4	SM (1)
Codon 12	Substitution	Gly12Asp	4	Non-SM
Codon 13	Substitution	Gly13Cys	2	SM (1)
Codon 12	Substitution	Gly12Ala	1	Non-SM
Codon 12	Substitution	Gly12Arg	1	Non-SM

*Abbreviations: n*+, number of patients with the same mutation in the lung tumor and the non–tumoral parenchyma (SM group). * One patient shared EGFR and KRAS mutations.

**Table 3 jcm-08-00529-t003:** Pattern of recurrence in mutated adenocarcinoma.

	SM	Non-SM	*p* Value
	*n* = 10	*n* = 37	
Recurrence, n (%) *	**6 (60)**	**3 (8.1)**	
Pattern of recurrence Local	0 (0)	1 (2.7)	*p* = 0.59
Distance	**6 (60)**	**2 (5.4)**	*p* < 0.001
Organ site of metastases			
Multiple organs affected	**3 (30)**	**0 (0)**	*p* = 0.001
Brain	**5 (50)**	**0 (0)**	*p* < 0.001
Hepatic	1 (10)	0 (0)	*p* = 0.55
Suprarenal	0 (0)	1 (2.7)	*p* = 0.59
Contralateral lung	1 (10)	0 (0)	*p* = 0.55
Bone	1 (10)	0 (0)	*p* = 0.55
Lymph node	**2 (10)**	**1 (2.7)**	*p* = 0.47

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
