# Peer review of "EGFR and KRAS Mutations in the Non-Tumoral Lung. Prognosis in Patients with Adenocarcinoma"

_jcm, 2019, doi:10.3390/jcm8040529_

Round 1

Reviewer 1 Report

The manuscript is describing the impact of mutations present in the "non-tumor" regions of the lung on to the progression of the cancer. It is an interesting research direction but its clinical relevance is not clear. as the authors pointed out, it is hard to believe that the area they call as tumor free is actually tumor free. Apart of cell free DNA and circulating tumor cells, there are extracellular vesicles that might be transferring the mutagenic load to near by regions. Although, the authors claim it will be ethical to sample another lung from humans, why not to recapitulate this in an animal model. The whole premises of having alteration or similarity in mutational profile fo tumor and non tumor regions is flawed as they are not able to prove that Non tumor region is really tumor free. 

The sample size is small and do not account for tumor micro-enviornment effect.  

Some work in cell lines and animal models is critically needed to make the conclusions believable.

Author Response

Please find attached a cover letter with the point-by-point comments.  

Reviewer 2 Report

1-    Very interesting study. However, the results are superficially presented and need significant improvement. Non-neoplastic tissues are removed from tissues adjacent to ADCs. According to the details information provided by the author, a minimum margin of 50px is considered. One major limitation of this procedure is the increased chance for sample cross contamination during surgical procedures. This needs to be clearly explained in the methodology section.

2- Digital PCR was performed to identify point mutations in all samples. No quantitative data was provided from neither the ADC nor the neoplastic samples.  This would be highly informative and strengthen the results. Also, the data could be compared with IHC results to assess whether the 2 experiments are consistent.

3- The presence of mutations in non-tumoral tissues implies 2 phenomena: (i) local dissemination of cancer cells from parental tumors, in which case both parental and normal-like tissues should have  a least one mutation in common; (ii) independent cells with driver mutations in which case the mutations may be different between the tumor and adjacent tissues. A pathological approach may not be the best strategy to evaluate the above possibilities. With regards to that, no strong evidence indicating non-tumoral tissues as the source of EGFR and KRAS mutations was provided. More direct evidence is needed (ei: Mutant Kras specific staining or FISH could be included with good negative controls).

4- Abbreviations such as SM are used throughout the manuscript without being clearly defined in the text.

5- Considering that a few patients have multiple mutations, it is important to evaluate how this could affect the recurrence or patient DFS rate. It is unclear whether the multiple mutations were identified in ADC or normal-like adjacent tissues. To assess possible connections between original tumor and adjacent tissues, a side-by-side comparison of identified mutations would be informative. Similarly, a graphical representation of EGFR/ KRAS exons and their mutation rate (table 2) would be appreciated by readers.

6-      Figure 1 needs a better labeling with clear indication of tumor location. The provided images do not allow a clear appreciation of the staining at the cellular level. This requires more magnification especially for areas with ADCs. Provide semi-quantitative data for mutant Kras staining. The fact that adjacent cells show positive staining suggests the need for a negative control. This will validate the specificity of the antibody used.

7-      Figure 2 has unreadable annotations that need to be fixed.

Author Response

Please find attached the cover letter with the point-by-point comments. 

Reviewer 3 Report

The authors reported their experience on the presence of EGFR and KRAS mutations in the NON-Tumoral Lung (NTL) of patients undergoing surgery for primary pulmonary adenocarcinoma.

The aim of the strudy was to analyze whether the most prevalent mutations observed in ADC (EGFR and KRAS) can also be observed in the non-neoplastic lung tissue, 

Moreover, they evaluated the short-term prognosis (1, 2, 6 and 12 months after surgery) of these findings.

169 patients were operated on in a 6-year period. EGFR and/or KRAS mutations were identified in 57 patients. The authors studied the NTL parenchyma specimens obtained during surgery from 47 patients with EGFR and/or KRAS mutations in their adenocarcinoma. 

The authors observed that the same mutations (SM) were present in the tumor and the histologically normal tissue in 21.3% of patients (SM group). Although local recurrences were similar in both groups, distant metastases were more frequent in the former (60 vs. 5.4%, p<0.001). Moreover, SM patients showed lower time-to-progression (8.5 vs. 11.7 months, p<0.001) and disease-free survival (8.5 vs. 28 11.2 months, p<0.001). COX regression showed a higher risk of progression or death (DFS) in the SM group (HR 5.94, p<0.01).

The authors concluded that genetic changes were present in the apparently normal lung in many adenocarcinoma patients, and this finding had significant prognostic implications.

The paper is well written and easy to follow. 

The topic is  interesting.

Some concerns should be addressed by the authors:

- Since the authors operated on 169 patients with primary pulmonary adenocarcinoma in a 6-year period, it seems to me it's not a high volume center. From this it follows that the same mutations  (tumour and NON-tumour tissue) were found only in 21.3% of the patients, that means 9 patients. This subgroup is quite low. These results need to be confirmed over a larger group of patients.

- As the authors pointed out, one year follow-up for patients undergoing surgery is extremely low.

The authors should look at the overall survival over a longer time period.

- pag 4, line 138: I guess the authors would have said "neo-adjuvant" instead of "adjuvant".

- Baseline characteristics between the two groups (SM and NON-SM) were significantly different in terms of smoking history, grade of COPD and aggressiveness of the tumour (PET uptake), all of them in favour of the NON-SM group. This could reasonably justified the results observed by the authors. Could the authors comment on that?

- Looking at table 2, 24 patients were found to have EGFR mutations. 27 EGFR mutations were reported. In the text (pag 5 line149) the authors said that "four patients carried two different mutations in their EGFR gene". So it doesn't add up.

- pag 7, line 178. The authors reported that "the brain was the most prevalent organ affected by distant metastases". Did the patients undergo a preoperative brain CT-scan? The rate of recurrence (local and distant) within such a short period of time is quite high (especially for patients undergoing surgery mostly having a pathological stage I disease). I was wondering whether the patients were accurately studied before surgery. Could the authors comment on that?

- The authors should clearly define "time to progression" and "disease free survival". It seems to me they are the same. Infact, looking at figures 2A and 2B, they seems to be overlapping. Could the authors comment on that? As I aforementioned, the authors should look at the overall survival.

- pag 10, lines 238-240. The authors reported that those patients with similar mutations in the tumor and the NTL parenchyma disclosed more metastasis and a lower DFS than those patients without. Therefore they suggested that "in the future, the use of a more aggressive approach in the follow up of these patients could be a valid attitude". I also would add that "the use of adjuvant therapy should be proposed to these subset of patients in order to prolong the DFS".

Author Response

(The authors gave the same response as above.)

Round 2

Reviewer 1 Report

The authors should update their discussion to reflect the reasons for not using animal models and cell lines and justifying their selection of tumor free regions. A description of clinical accepted definition of tumor free regions is also needed in the manuscript. 

The small sample size is still a concern to evaluate the statistical significance of their findings. These concerns should included in the discussion. 

Author Response

In the document attached below, please find the answer to your comments in a point-by-point format.

Reviewer 2 Report

I thank the authors for providing more specific details. The new version allows better understanding of the manuscript. The proposed graphical representation should be included  and the IHC staining images (figure 1) need scale bars.

I still think that representative images of EGFR and KRAS staining ( at least the total proteins) need to be added to the manuscript to give a sense of their expression level. H&E staining is less important as it does not add much to your hypothesis.

Author Response

(The authors gave the same response as above.)
